# Investigation of the Impact of High-Speed Machining in the Milling Process of Titanium Alloy on Tool Wear, Surface Layer Properties, and Fatigue Life of the Machined Object

**DOI:** 10.3390/ma16155361

**Published:** 2023-07-30

**Authors:** Jakub Matuszak, Kazimierz Zaleski, Andrzej Zyśko

**Affiliations:** Department of Production Engineering, Mechanical Engineering Faculty, Lublin University of Technology, Nadbystrzycka 38D, 20-618 Lublin, Poland; k.zaleski@pollub.pl (K.Z.); andrzej.zysko@wp.pl (A.Z.)

**Keywords:** surface layer, high speed machining, Ti-6Al-4V titanium alloy, residual stress, microhardness, fatigue life, tool wear, milling

## Abstract

This article presents the results of experimental research on the effect of high-speed machining (HSM) in the milling process on the tool wear, surface layer properties, and fatigue life of objects made of Ti-6Al-4V titanium alloy. Titanium alloys are widely used in many industries due to their high strength-to-density ratio, corrosion resistance, and resistance to dynamic loads. The experiment was conducted on a vertical three-axis machining centre, Avia VMC800HS. The influence of increased cutting speeds on the average values and amplitudes of the total cutting force components and the surface roughness of the machined workpiece was determined. Variable cutting speeds v_c_ = 70; 130; 190; 250; 310 m/min were applied. The impact of HSM on machinability indicators, such as the microhardness of the surface layer, the distribution of residual stresses, and the fatigue life of the samples after milling, was analysed. The thickness of the hardened layer varied from 20 to 28 micrometres. The maximum compressive residual stress Ϭ_m_ = 190 MPa was achieved at the speed of v_c_ = 190 m/min. A significant influence of increased cutting speeds on tool wear was demonstrated. The longest tool life (t = 70 min) was obtained for low cutting speeds (conventional) v_c_ = 70 m/min.

## 1. Introduction

Titanium alloys are used in many industries where it is essential to maintain strength parameters while maintaining low structural weight, especially in the aerospace and automotive industries, primarily due to their high strength-to-density ratio, corrosion resistance, and resistance to dynamic loads. The Ti6Al4V alloy, also known as Titanium Grade 5, is one of the most commonly used titanium alloys. It shows very good corrosion resistance in various environments, including seawater and organic acids. However, due to its low thermal conductivity and high hardness, titanium alloys belong to the group of difficult-to-cut materials. The accumulation of heat generated during plastic deformation in the chip formation zone directly in the contact area with the cutting tool negatively affects the tool. The exceeded critical temperature for a specific type of tool material results in the occurrence of processes that contribute to accelerated tool wear.

The key factors that determine both the surface quality of a machined component and the technological efficiency of the cutting process are the tools, more specifically, their design features and the material of the cutting tool blade.

The most commonly used cutting tools in the machining of titanium and its alloys, due to their acceptable combination of hardness and strength, are the tools made of cemented carbides. They have high hardness and wear resistance, enabling a 3–4 times increase in cutting speeds compared to high-speed steel tools.

Blade wear in the milling of titanium alloys occurs as a result of adhesive action together with the intensive abrasive action. The resistance of carbide tools to this type of wear can be improved by reducing the grain size. The life of end mills with an ultrafine-grained structure is several times longer than those with a coarse-grained structure [1,2]. 

Tool wear can take many forms: mechanical, thermal, diffusion, or chemical [3]. The gradual abrasive wear occurring on the rake and flank surfaces is an inevitable part of the cutting process, especially when machining difficult-to-machine materials. A negative phenomenon is catastrophic wear because it is more difficult to predict compared to abrasive wear. There are several causes of catastrophic wear: too high cutting parameters, increasing abrasive wear, which leads to an increase in cutting forces and excessive tool load, and exceeding the permissible cutting temperature. The occurrence of catastrophic wear is associated with the need to stop machining on CNC machines and to replace tools; sometimes, it can contribute to damage to the workpiece. For this reason, it is important to analyse different forms of wear [4]. Samborski et al. [5] focused on evaluating the fracture toughness of WC cutting inserts, specifically the SM25T (HW) tungsten carbide. They applied fracture mechanics methods, including quasi-static three-point bending tests and Charpy impact tests, to calculate the static and dynamic fracture toughness. The microscopic analysis of fracture surfaces was also conducted to examine material irregularities. The authors concluded that their proposed examination solution can successfully determine the toughness properties of WC-based materials and demonstrated agreement with other works.

The introduction of anti-wear coatings in the milling process of titanium alloys makes it possible to increase machining efficiency, reduce the frequency of tool changes, and obtain a better quality of the machined surface. Therefore, tools with anti-wear coatings are widely used because they contribute to increasing the tool life. The coatings are designed to protect the tool tip against the intense adhesive and abrasive effects of the workpiece. The anti-wear coatings can also reduce friction and adhesion phenomena between the tool tip and the workpiece. Thanks to this, the accumulation of heat in the tool is limited, which is important when machining titanium alloys that have low thermal conductivity. The reduction of the friction coefficient of working surfaces of tools, through the use of protective coatings, also helps to prevent chips from sticking to the tool, which can lead to damage and increased wear on the cutting edge.

Despite the protective properties of tool coatings, the research has shown that uncoated cemented carbide (WC-Co) tools show better machining results when machining titanium alloys compared to tools with TiC, TiN-TiC coatings [6]. Lebaal et al. [7] showed that the wear of the blades of the cemented carbide inserts by means of which the Ti-6242S alloy was milled is not significantly dependent on the coating of the cutting insert but depends on the technological parameters of machining. The blade wear mechanism was also analysed during the milling of Ti-6242S titanium alloy with uncoated and TiN-, TiC-, and TiCN-coated carbide cutters [8]. It was shown that one of the causes of tool wear is the thermal load on the cutting edge to which it is subjected during machining.

Proper cooling while milling titanium alloys is crucial for several reasons. Firstly, titanium has poor thermal conductivity, which means that it has a low ability to dissipate heat generated during machining. This can lead to excessive heat build-up in the cutting zone, causing tool wear and reduced tool life. Additionally, cooling helps to prevent the adhesion of chips to the cutting tool; this can cause built-up edge (BUE) formation and affect the cutting performance. The use of coolants or cutting fluids helps to reduce friction and facilitates chip evacuation, thereby improving cutting efficiency and surface finish. Moreover, cooling can minimize the occurrence of thermal stresses and distortions in the workpiece, ensuring dimensional accuracy and reducing the risk of part deformation. The selection of the appropriate cooling method, such as MQL (minimum quantity lubrication) or high-pressure coolant systems, can significantly enhance the machining process of titanium alloys by reducing friction, cutting forces, and temperature, thus improving tool life, surface quality, and overall productivity [9,10]. Reduction in tool wear through the use of cutting fluids can range from 20% to 50% [11]. 

Khatri et al. [12] analysed the effect of various cooling methods on tool wear during the slot milling of Ti6Al4V titanium alloy. The study showed that dry machining with TiAlN-coated tools and MQL machining resulted in less tool wear compared to liquid machining.

The use of a high-pressure coolant system has a positive effect on both the tool life and the machined surface quality. High pressure leads to improved chip evacuation and faster heat dissipation from the cutting zone. This type of cooling reduces the friction coefficient and thus cutting force and improves chip fragmentation [13,14].

Kulianic et al. [15] concluded that the most suitable type of conventional cooling for milling titanium alloy with PCD tools provides a coolant containing approximately 7% oil and water. Kitagawa et al. [16] demonstrated that high-speed milling (HSM) with carbide end mills is feasible only when abundant cooling is applied.

Machining titanium alloys in an argon-enriched environment provides a longer tool life compared to conventional cooling methods [17]. 

The theoretical foundations of HSM machining compared to conventional machining assume the occurrence of lower cutting forces [18,19]. As cutting speed increases, cutting forces decrease by 10–15% [20]. This can have a positive effect on blade life [21,22]. Many research centres conduct research to verify these assumptions for various materials, including titanium alloys. However, for each group of construction materials, the speed range of the HSM process is different. During the milling of titanium alloy Ti-6Al-4V, Vijay et al. [23] demonstrated that increasing the cutting speed from 30 m/min to 40 m/min resulted in a decrease in the force of approximately 10%. However, further increasing the speed to 60 m/min did not cause any change in its value.

The research on the influence of cutting speed in the milling of thin-walled components on cutting force has shown that within the range of cutting speeds from 60 m/min to 120 m/min, the cutting force increased. It decreased in the range of (120 ÷ 150) m/min, and there was a subsequent increase in this force in the range of (150 ÷ 180) m/m [24]. The analysis of the course of cutting forces is a valuable factor in the machining process: It allows for evaluating the need to replace a tool, locating discontinuities and defects in workpieces [25,26,27,28], and identifying areas of unstable machining while cutting the thin-walled components [29,30,31,32]. 

During milling, large amounts of heat are generated, especially in the contact zone of the tool with the workpiece. Depending on the distance from the contact zone, the heating rate changes. The thermal properties of titanium alloys may change in response to varying machining conditions. In addition, the composition of the titanium alloy and the heating rate affect the start and end points of the phase transition [33,34].

Due to the problems with machining titanium alloys, attempts are made toward multi-criteria optimisation, taking into account the analysis of surface roughness, force, and temperature in order to determine the optimal cutting parameters: cutting speed and feed [35].

Surface roughness is a key parameter in the case of milling titanium alloys due to their difficult-to-cut properties and dynamically increasing wear processes, which translates into changes in surface roughness [36]. In order to minimise roughness, it is necessary to properly select cutting parameters, such as cutting speed and feed, but it is also important to take into account the tool geometry and milling strategy [37]. In addition, the use of appropriate cooling and lubrication technologies can help improve the surface quality of workpieces made of titanium alloys. 

Ginting et al. [38] analysed the influence of anti-wear coatings on surface roughness. It was demonstrated that lower surface roughness values were observed after dry milling using uncoated tools compared to coated tools made of cemented carbide. It was concluded that this can be achieved in both roughing and finishing operations.

Su et al. [39] investigated the effect of tools made of advanced materials, such as polycrystalline diamond (PCD) and polycrystalline cubic boron nitride (PCBN), on surface roughness during high-speed milling of Ti-6.5Al-2Zr-1Mo-1V (TA15) alloy. They found that the PCD tool had a significantly longer tool life compared to the PCBN tool, especially at higher cutting speeds. It was demonstrated that high-speed milling of titanium alloy can be performed while maintaining low surface roughness values below 0.3 µm.

The specificity of machining titanium alloys makes the process conditions strongly affect the surface layer properties. Adjusting the cutting parameters can affect the microhardness by controlling the deformation zone in the material. It was demonstrated that with an increase in cutting speed, the values of compressive residual stresses increase [40]. However, with an increase in feed, the compressive residual stresses decrease. 

The issues related to the machining of titanium alloys are intensively studied in many research centres due to the difficulties with the machining of these materials. The research focuses on the analysis of tool wear and machined surface quality, as well as the surface layer properties, such as microhardness or residual stresses. However, studies on the impact of HSM (high-speed machining) on these aspects are particularly interesting because they bring original conclusions regarding the machining of difficult-to-cut materials.

## 2. Materials and Methods

### 2.1. Shape of Research Material

Titanium alloy Ti 6Al-4V (Grade 5) is one of the most commonly used titanium alloys. This alloy is used to produce many parts for which machining operations are carried out, such as turbine blades, aircraft structural components, sports equipment, valves, and pumps. Relatively large, rigid samples of Ti-6Al-4V titanium alloy in the shape of a cuboid with dimensions of 150 mm × 90 mm × 40 mm were used to study the cutting torque and thrust force as well as surface roughness in order to avoid vibration and ensure rigid mounting on the CNC machine. On the other hand, for the measurement of residual stresses and also for fatigue life testing, specimens with shapes matching the design of the authors’ test stands were used. Figure 1 shows the shapes and dimensions of the samples used. 

The chemical composition and physical properties of the tested material are presented in Table 1.

### 2.2. General Methodology

Schematic diagram of this study is shown in Figure 2. The tests were conducted on the Avia VMC800HS three-axis vertical machining centre. For the milling process, the Seco 20 mm diameter double-blade milling cutter (symbol: R217.21-1820.0-LP06.2A) with replaceable carbide inserts (symbol: LPHT060310ER-E05) and grade MM4500 was used. Variable cutting speed v_c_ = 70; 130; 190; 250; 310 m/min and fixed parameters: depth of cut a_p_ = 0.5 mm, milling width a_e_ = 7.5 mm, feed per tooth f_z_ = 0.2 mm/tooth were applied during the experiment.

The general experimental methodology is shown in Figure 3. The research focused on the analysis of mechanical wear. The blade wear indicators were determined based on the ISO 8688-2: 1996 standard.

An exaggerated and localized form of flank wear which develops at a specific part of the flank called “localized flank wear: VB3” was adopted as a wear indicator in this research. According to the standard, a tool is worn if the VB3 indicator value exceeds 0.5 mm.

This experiment was repeated eight times for five different cutting speeds. For each repetition, the flank wear (VB3) was recorded until the total wear of the blade VB3 = 0.5 mm.

### 2.3. Measurements Equipment 

The wear rates were measured using the Keyence (Japan) digital microscope model VHX 5000. Measurements of thrust force and cutting torque were carried out on the Avia VMC800HS vertical 3-axis milling centre equipped with the Kistler 9125A force gauge and the 5070 type amplifier. The measured signals were recorded using Dynowave software (type 2825A) with a Dynoware data acquisition card (type 5697A). From the stable course of force (ignoring the unstable zones of the tool entry and exit from the material), the average values of thrust force and cutting torque were determined. An example course of the thrust force during milling at v_c_ = 130 m/min is shown in Figure 4.

The surface topography was measured using the T8000RC120-400 profilographometer provided by Hommel-Etamic Jenooptik (Jena, Villingen-Schwenningen, Germany). The surface topography was tested on the 4.8 mm × 4.8 mm surface using the contact method after milling with variable cutting speed. Roughness parameters, such as Sa (arithmetical mean height) and Sz (maximum height), were measured. Eight samples were made for each of the five cutting speeds. Roughness measurements were carried out in the direction perpendicular to the traces formed after the milling operation. Based on the measured values, the average and standard deviation of the obtained measurements were determined. 

The surface microhardness was determined with the Leco LM700 device in compliance with the EN-ISO 6507-1:2018 standard. The Leco tester was fitted out with a 10× magnification eyepiece and a 50× magnification lens. The penetrator loading time was 15 s with the 50 g load applied. 

The tests of residual stress distribution were carried out using the mechanical method. The residual stress measurements were carried out on the original test stand for chemical etching with mechanical removal of etching products. The titanium alloy was etched in a 4% aqueous solution of hydrofluoric acid. Residual stresses were determined by measuring the size of sample deformation caused by the removal of material layers in which residual stresses persist. The diagram of the stand for measuring residual stresses is shown in Figure 5. The test stand works according to the principle that the sample layers with residual stresses are removed under the action of acid. The sample is fixed in sliding supports, which ensures its free deformation when removing layers with residual stresses. The sample deflection is measured during the test. The residual stress is determined on the basis of the sample deflection.

The cutting speed can affect changes in the surface layer and, as a result, the fatigue life. The fatigue life was analysed on the original test stand shown in Figure 6

## 3. Results

### 3.1. Tool Wear Tests

Tool wear is one of the most important issues in the milling process. It affects both the unit cost of part production and the final quality of the manufactured product. Figure 7 shows the wear increment (VB3) of the example blade on the flank surface for a cutting speed of v_c_ = 70 m/min.

For the machining of materials such as titanium alloys, the cutting speed is one of the key parameters affecting the tool blade life in the cutting process. The aim of this research was to analyse the milling of titanium alloy for variable cutting speeds in order to determine the blade life for individual v_c_ with constant technological parameters: f_z_, a_e_, a_p_. In addition, the maximum values (v_c_) for the milling of titanium alloy under HSM conditions were determined. The blade durability was compared under conditions of increased cutting speeds as well as conventional machining. The selected values (v_c_) were applied during the further stages of the research.

Figure 8 shows a summary of blade wear as a function of time for the tested cutting speeds: v_c_ = 70, 130, 190, 250, 310 m/min.

Upon analysing the conducted tests, it can be concluded that for the machining of titanium alloy Ti-6Al-4V with carbide insert tools, the longest tool life was obtained for the low cutting speeds (conventional), v_c_ = 70 m/min. Milling at speeds of v_c_ = 250 m/min and v_c_ = 310 m/min causes a significant development of tool wear in the initial phase of cutting. The value of the VB3 coefficient specified in the ISO 8688-2: 1996 standard was exceeded after 1 min (for the speed v_c_ = 250 mm) and after about t = 0.5 mm (for v_c_ = 310 m/min). Therefore, in an industrial application, these speeds are not recommended (even if they introduce beneficial changes in surface layer properties).

### 3.2. Tests of Thrust Force and Cutting Torque in the Milling Process

An increase in blade wear during machining causes an increase in self-excited vibration and thrust force; in addition, an increase in cutting speed deteriorates the condition of the cutting edge and intensifies this process. Therefore, it is important to know the range of favourable machining conditions to reduce components of the total cutting force. Figure 9 shows the effect of cutting speed on the average value of thrust force. In the range from 70 m/min to 250 m/min, the thrust force took on similar values. There was a noticeable increase in force at 310 m/min, which may have been caused by unstable cutting conditions due to accelerated wear.

Figure 10 shows the effect of cutting speed on the value of cutting torque. No significant effect of cutting speed on cutting torque was observed. In the entire range of cutting speeds adopted in the experiment, the cutting torque was in the range from 0.9 to 1 Nm.

### 3.3. Surface Roughness

The main factors that may affect the surface quality after milling include the cutting parameters and the cutting edge condition. Their correct selection and the monitoring of the cutting edge condition during machining ensure the reduction of roughness parameters of the machined surface. Table 2 presents example surface topographies after milling with variable cutting speed periodicity characteristic of milling operations. The periodicity of the shaped surface has a determinate form. Peaks characteristic of the milling process were observed on the surface, occurring at constant distances resulting from the feed per tooth and the mapping of successive traces of tool blades in the workpiece. 

The average values of the Sa parameter are presented in the graph of the impact of cutting speed on the surface topography (Figure 11). In the speed range from 70 to 190 m/min, a decrease in surface roughness is noted as the speed increases, which is consistent with the HSM process. However, for speeds of 250 and 310, an increase in surface roughness is evident.

With the increase in cutting speed up to 190 m/min, the temperature increases, which causes heating of the material and a decrease in the friction coefficient, while above this speed, rapid edge wear occurs (Figure 8), which may be the dominant factor influencing the increase in surface roughness.

### 3.4. Surface Layer Microhardness

Figure 12 shows the microhardness distribution of the surface layer of samples milled at the speed of v_c_ = 70 m/min. A characteristic increase in the surface layer microhardness was observed just below the surface. 

Figure 13 shows the effect of cutting speed on the thickness of hardened layer gh. The thickness of the hardened layer varied from 20 to 28 micrometres. The decrease in thickness of the hardened layer at a speed of v_c_ = 250 and v_c_ 310 m/min could be caused by accelerated tool wear and the temperature increase.

### 3.5. Residual Stress

Figure 14 shows an example of stress distribution in the surface layer after milling at 130 m/min. The diagram shows typical indices as:Compressive residual stress on the surface Ϭ_p_;Maximum compressive residual stress Ϭ_m_;Depth of maximum compressive residual stress g_m_;Depth of total compressive residual stress g_s_.

Generation of compressive stresses during milling can be a positive phenomenon Compressive residual stresses can contribute to improving material strength, cracking and fatigue resistance, and reducing the tendency to corrosion. 

Figure 15 and Figure 16 show the impact of cutting speed on the values of maximum compressive residual stress and the depth of total compressive residual stress, respectively.

Up to a speed of 190 m/min, the maximum compressive residual stress increases with increasing speed, as does the depth of total compressive residual stress. Then, a decrease in the value of tested indicators can be observed with the increase in the cutting speed. This may be related to the temperature effect at increased cutting speeds (where tool wear processes also develop faster).

### 3.6. Fatigue Life Tests after Milling

Each type of machining (milling, turning, grinding) affects the surface layer condition, and subsequently, this has an impact on the fatigue life of the material. The high surface roughness of machine elements is one of the factors influencing the formation of fatigue cracks. In the case of milling, surface irregularities resulting from the passage of the cutting edge with the adopted feed f_z_ can be treated as micro-notches. Notches appearing on a rough surface cause an increase in stress concentration and strain, which significantly reduce the fatigue properties of the material. The factors significantly influencing the initiation and further propagation of fatigue cracks are both residual stresses and material hardening. Hardening of the surface layer inhibits the initiation of fatigue cracks but accelerates their propagation. On the other hand, compressive residual stresses inhibit stress propagation to a greater extent. Tensile residual stresses are negative both in terms of the initial phase (initiation) of fatigue crack formation and their further development (propagation). 

Both the cutting tool edge wear and the cutting speed affect the above factors and the surface condition and, at the same time, the fatigue strength of manufactured components.

Figure 17 shows the effect of cutting speed on fatigue life after milling. A trend of increase in fatigue life can be observed with increasing speeds up to 190 m/min followed by a slight decrease in fatigue life for speeds of 250 and 310 m/min.

## 4. Conclusions

This article presents an analysis of the impact of cutting speed on the surface layer properties and fatigue life of the Ti6Al4V titanium alloy after milling. The tests were carried out in the range of conventional speeds as well as in high-speed machining (HSM) conditions. Among the many different types of wear of the cutting tool edge, flank wear (VB3) was selected. Due to its properties, Ti6Al4V alloy is a frequently used alloy in aerospace structures. The large dimensions and the degree of complexity of the parts characteristic of this branch of industry sometimes make the milling process the only and final treatment occurring in the process of their production. One of the methods of optimising the process of milling parts from titanium alloys in relation to conventional milling is HSM. Despite many advantages of HSM, excessively high cutting speeds for machining materials such as titanium alloys can lead to blade failure. Since the machining time for complex parts is relatively long, in order to maintain the required surface quality throughout the milling process, it is important to constantly monitor the tool’s cutting-edge condition. Milling with a tool in which wear has significantly increased can negatively affect the surface layer condition. Therefore, it is important to know how tool wear affects the condition of the surface layers of workpieces machined at high cutting speeds. The following results summarize this study of the effect of cutting speed during milling on the tool wear, surface layer properties, and fatigue life of titanium alloy Ti6Al4V:Machining of titanium alloy at higher cutting speeds reduces blade life;The best ratio of tool life to machining time under higher cutting speeds was obtained for the speed of 130 m/min;An increase in the cutting speed above v_c_ = 190 m/min results in a several times faster increase in the flank wear (VB3) compared to conventional speeds (v_c_ = 70 m/min);Cutting speeds in the range of v_c_ = 250 to 310 m/min are not recommended due to rapid tool wear: Despite favourable changes in the surface layer properties in industrial practice, frequent tool changes preclude the use of v_c_ = 250 and v_c_ = 310 m/min speeds in the production of parts made of Ti6-Al-4V titanium alloy.In the range of the adopted cutting speeds, no significant changes in thrust force and cutting torque were observed;Up to the cutting speed of v_c_ = 190 m/min, as the cutting speed increases, the surface roughness (Sa) decreases, and a further increase in cutting speed causes an increase in surface roughness: This may be related to the combination of several phenomena occurring in the process that with the increase in cutting speed up to 190 m/min, the temperature increases, which causes heating of the material and a decrease in the friction coefficient, while above this speed, rapid edge wear occurs (Figure 8), which may be the dominant factor;The cutting speed affects the hardened surface layer thickness: The thickness of the hardened layer varied from 20 to 28 micrometres;The cutting speed affects the residual stresses: The highest values of maximum compressive residual stress σ_m_ and the largest thickness of total compressive residual stress g_s_ were obtained for the speed of v_c_ = 190 m/min;Changes in the properties of the surface layer caused by milling affect the fatigue life, which is an important factor in terms of variable loads of manufactured elements: The largest number of cycles was obtained for the speed v_c_ = 190 m/min.

## Figures and Tables

**Figure 1 materials-16-05361-f001:**
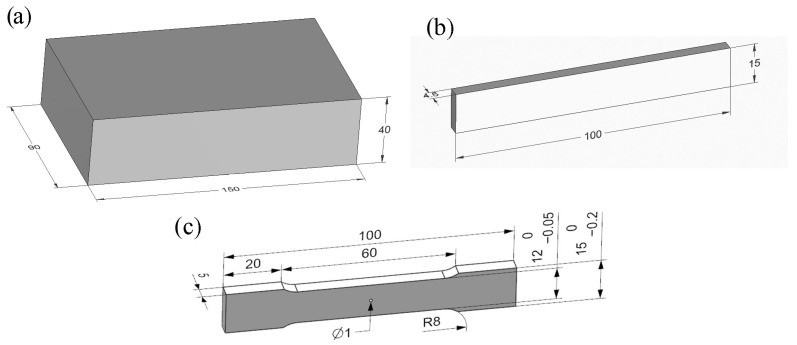
Shape and dimensions (in millimetres) of samples depending on the type of experiment: (**a**) force, torque, surface roughness, (**b**) residual stress, microhardness, (**c**) fatigue life.

**Figure 2 materials-16-05361-f002:**
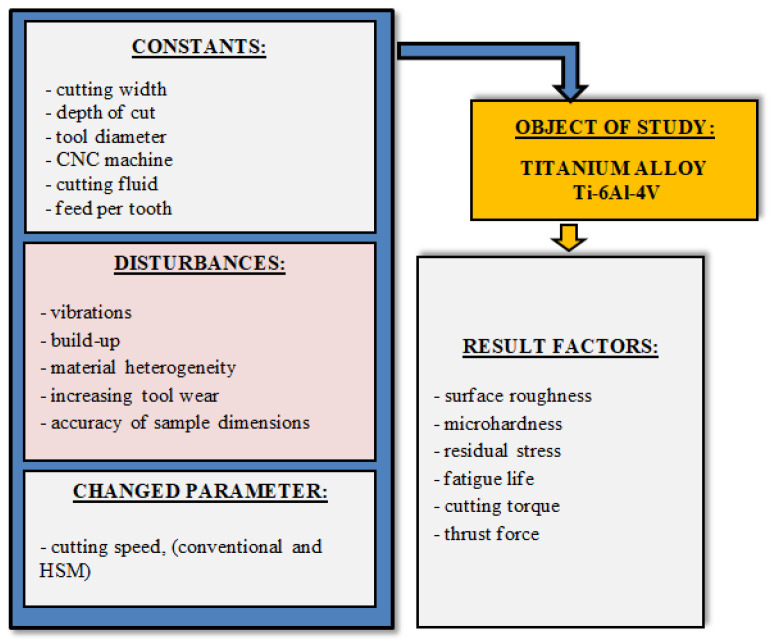
Schematic diagram of the study object.

**Figure 3 materials-16-05361-f003:**
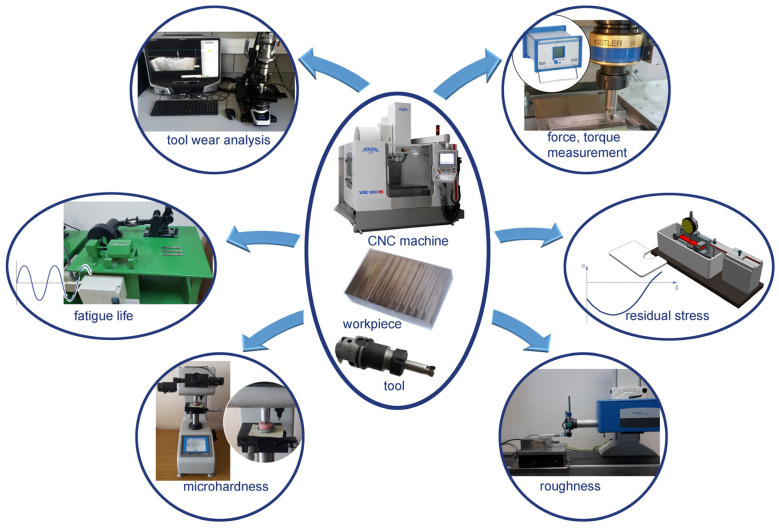
General diagram of the experiment methodology.

**Figure 4 materials-16-05361-f004:**
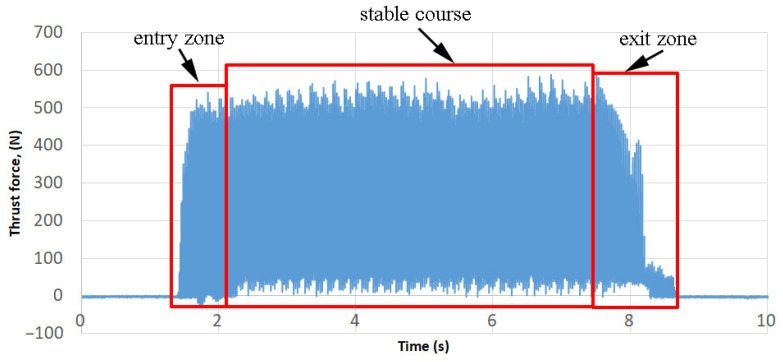
Example of the thrust force during cutting at v_c_ = 130 m/min.

**Figure 5 materials-16-05361-f005:**
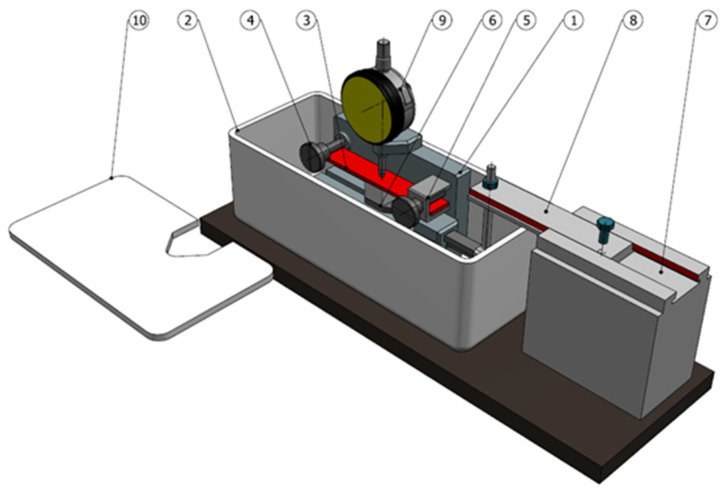
Model of the test stand for measuring residual stresses: 1—support body; 2—acid container; 3—test sample; 4—fixed support; 5—sliding support; 6—contact brush; 7—brush drive; 8—linear guide; 9—digital sensor; 10—container lid.

**Figure 6 materials-16-05361-f006:**
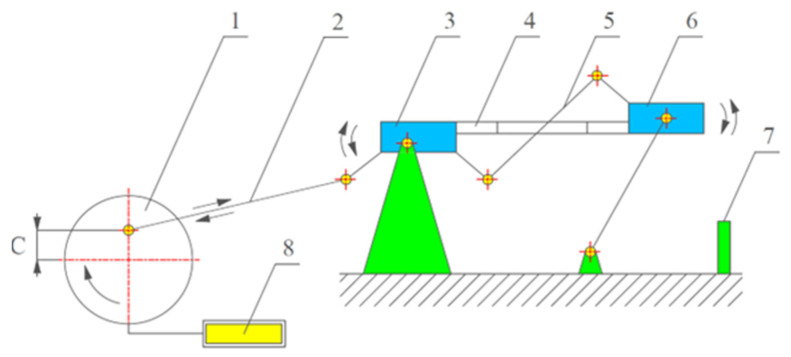
Test stand for measuring fatigue life: 1—drive shaft; 2—connecting rod; 3—left mounting bracket; 4—test sample; 5—lever system; 6—right mounting bracket; 7—safety switch; 8—cycle counter.

**Figure 7 materials-16-05361-f007:**
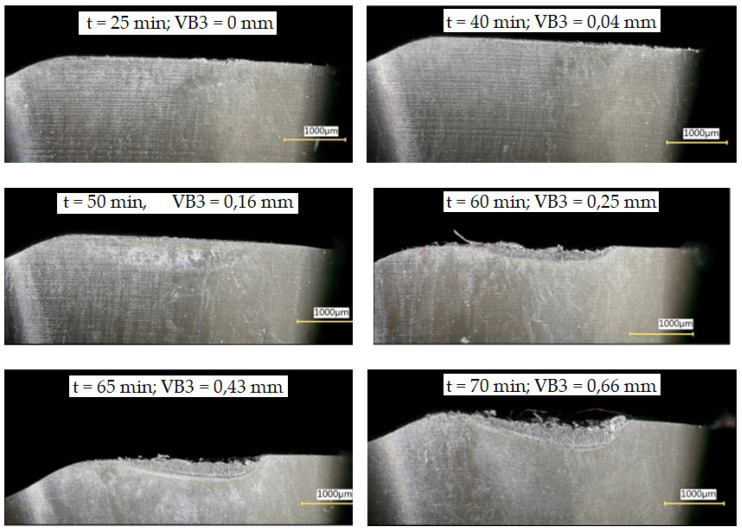
Development of flank wear (VB3) of the selected blade for cutting speed vc = 70 m/min.

**Figure 8 materials-16-05361-f008:**
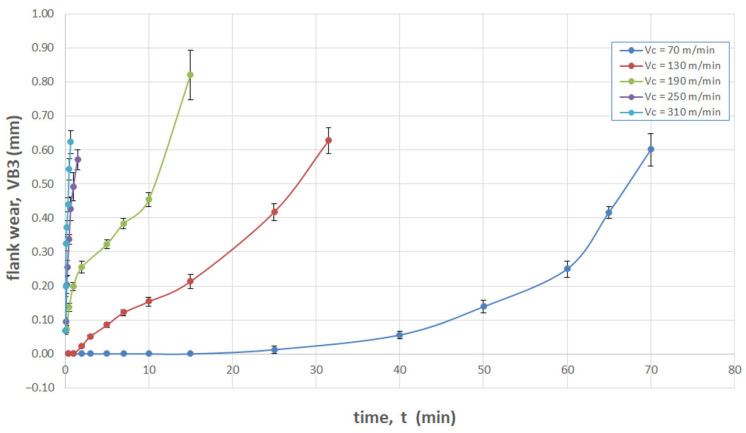
Blade wear for the adopted cutting speeds: v_c_ = 70 m/min, v_c_ = 130 m/min, v_c_ = 190 m/min, v_c_ = 250 m/min, v_c_ = 310 m/min.

**Figure 9 materials-16-05361-f009:**
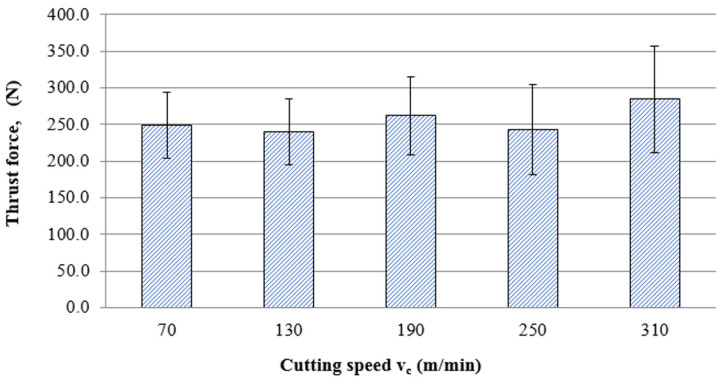
Effect of cutting speed on the average value of thrust force.

**Figure 10 materials-16-05361-f010:**
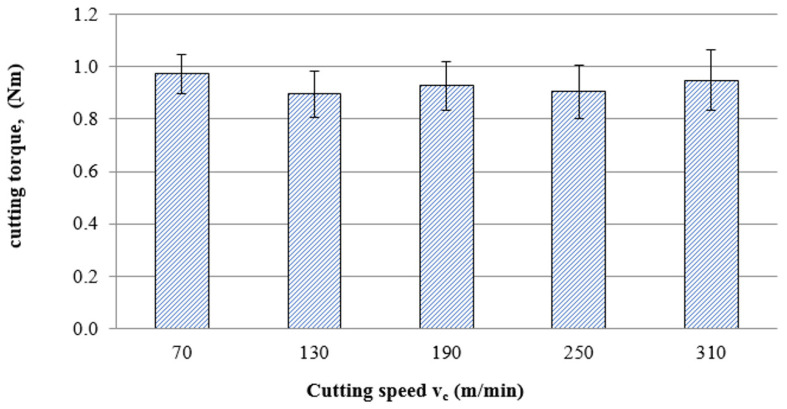
Effect of cutting speed on the value of cutting torque.

**Figure 11 materials-16-05361-f011:**
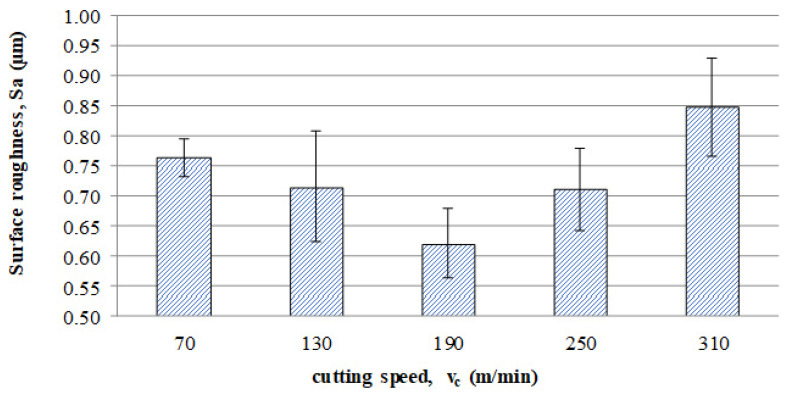
Influence of cutting speed on the Sa surface roughness.

**Figure 12 materials-16-05361-f012:**
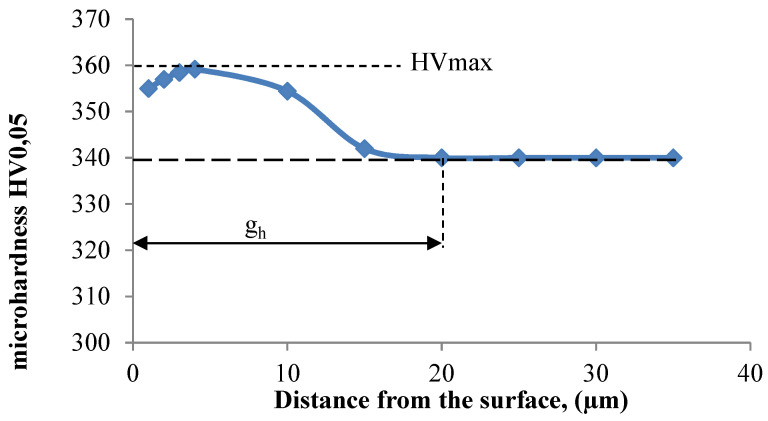
Distribution of surface layer microhardness as a function of distance from the surface after milling at the speed of v_c_ = 70 m/min.

**Figure 13 materials-16-05361-f013:**
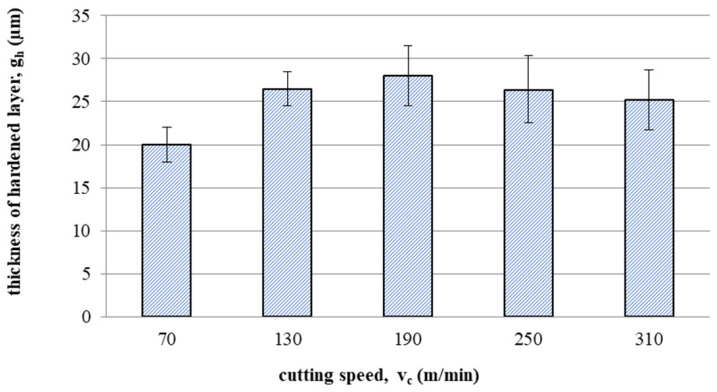
Effect of cutting speed on the thickness of hardened layer g_h_.

**Figure 14 materials-16-05361-f014:**
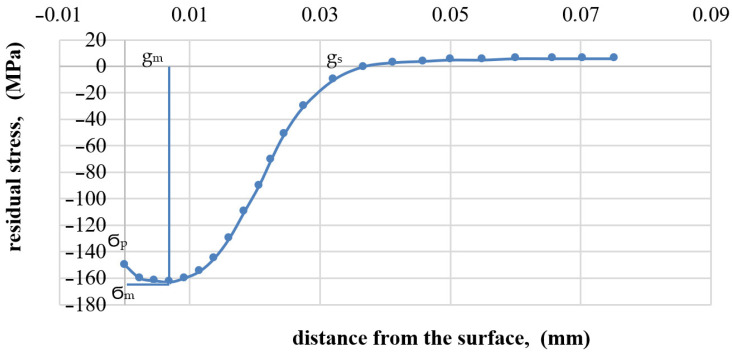
Residual stress distribution as a function of distance from the surface after milling Ti6Al4V titanium alloy at 130 m/min.

**Figure 15 materials-16-05361-f015:**
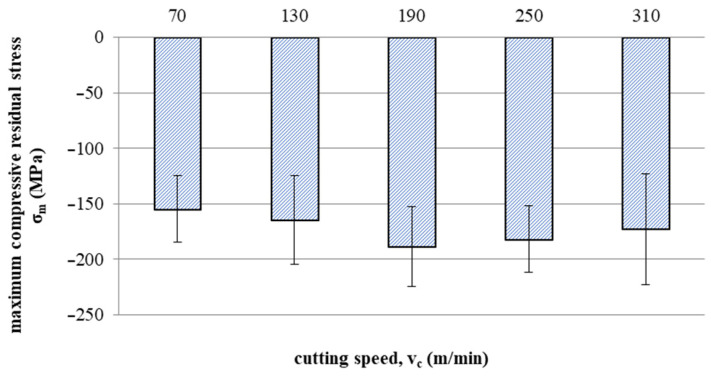
Effect of cutting speed on maximum compressive residual stress σ_m_.

**Figure 16 materials-16-05361-f016:**
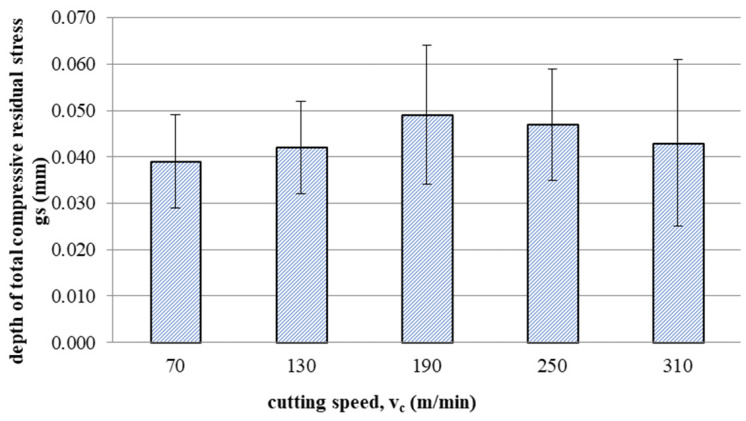
Effect of cutting speed on depth of total compressive residual stress g_s_.

**Figure 17 materials-16-05361-f017:**
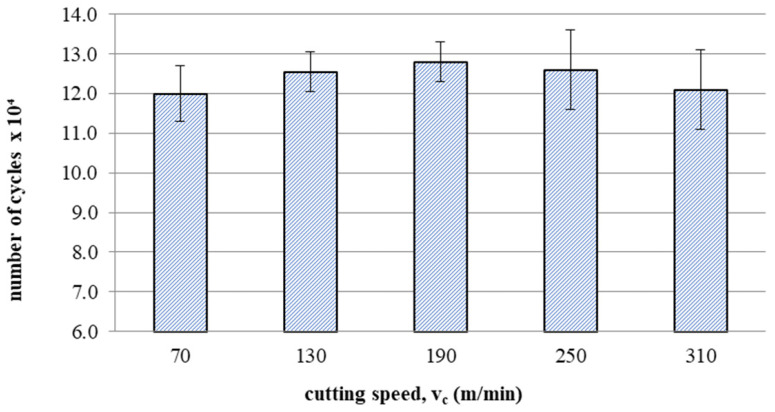
Influence of cutting speed on fatigue life after milling.

**Table 1 materials-16-05361-t001:** Chemical composition and physical properties of Ti-6Al-4V titanium alloy.

Chemical Composition, wt.%		Mechanical Properties	
Al	6.25–6.31	Rm (MPa) E (GPa)	1014 120
V	4.09–4.12		
C	0.026–0.027		
Fe	0.18–0.21	HRC	33
Ti	Rest		

**Table 2 materials-16-05361-t002:** Surface topographies after milling Ti-6Al-4V titanium alloy with variable cutting speed.

v_c_	Surface Topography	Roughness Parameters, (µm)
70 m/min	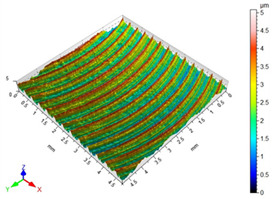	Sp = 3.47 Sv = 3.76 Sz = 5.23 Sa = 0.80
130 m/min	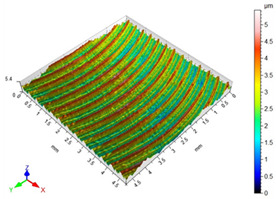	Sp = 2.49 Sv = 2.56 Sz = 4.95 Sa = 0.67
190 m/min	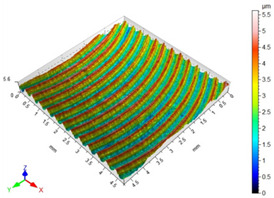	Sp = 3.47 Sv = 2.99 Sz = 5.45 Sa = 0.63
250 m/min	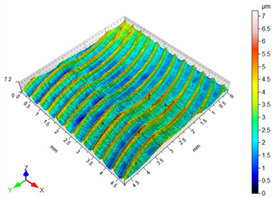	Sp = 3.08 Sv = 3.64 Sz = 7.12 Sa = 0.77
310 m/min	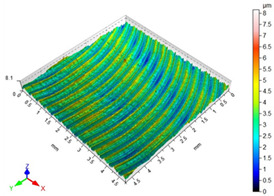	Sp = 3.05 Sv = 4.00 Sz = 8.08 Sa = 0.93

## Data Availability

Not applicable.

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
