# Peer review of "Investigation of the Impact of High-Speed Machining in the Milling Process of Titanium Alloy on Tool Wear, Surface Layer Properties, and Fatigue Life of the Machined Object"

_materials, 2023, doi:10.3390/ma16155361_

Round 1

Reviewer 1 Report

The article is devoted to the topical and important topic of mechanical processing of titanium alloys. The article should be finalized before publication.

The introduction is too big. It would be necessary to shorten the introductory, general part a little. It is also not necessary to describe the work of the authors in such detail. You can limit yourself to key things.

In the methodology section, it would be good to indicate why you chose this particular material for testing.

It would also be nice to divide the methodology section into subsections: Carrying out mechanical tests, Sample processing, Measurement of surface parameters, ...

Table 3 is a set of drawings and should be formatted as Figure 3 a, b, c, ...

In the figure you provide wear data. What is the spread of the obtained experimental values? Specify the confidence level.

In figures 6 and 7, the confidence probability is quite large, so it is quite difficult to talk about an increase or decrease in cutting forces. When describing, one should be more careful with this.

It would also be nice to have a small diagram in the methodology section showing the measured cutting force acting on the cutting insert.

You have a large amount of experimental material but little discussion of the results obtained. It would be good to describe in more detail what the resulting results are connected with, to provide data on the interaction of the tool material with the workpiece being processed.

Author Response

Article: Investigation of the Impact of High-Speed Machining in the Milling Process of Titanium Alloy on Tool Wear, Surface Layer Properties, and Fatigue Life of the Machined Object

Responses to comments from Reviewer 1

I would like to thank the Editor for their consideration, and the Reviewer for the time spent on carefully reviewing this work and for their valuable deep insight and comments. I feel that this paper is now clearer, more thoroughly discussed and better-referenced.

The work has been revised to address the reviewers’ suggestions. Please find hereafter a  point-by-point reply to the comments and suggestions. Any revisions to the manuscript was marked up using the “Track Changes”.

General comment from Reviewer 1:  The article is devoted to the topical and important topic of mechanical processing of titanium alloys. The article should be finalized before publication.

Response:

We would like to thank the Reviewer for his very good opinion. We feel to be obligated to answer for points mention in the review. The paper has been modified and improved. We believe that now is clearer.

Comment 1:  The introduction is too big. It would be necessary to shorten the introductory, general part a little. It is also not necessary to describe the work of the authors in such detail. You can limit yourself to key things.

Response:

Thank you for your valuable feedback. We have taken your suggestions into consideration and made the necessary changes to the introduction. We have shortened the introductory and general part while ensuring that the key points are still adequately addressed. Additionally, we have reduced the level of detail regarding the authors' work, focusing only on the essential aspects.

Comment 2: In the methodology section, it would be good to indicate why you chose this particular material for testing.

Response:

This is a very valid point. We have addressed your suggestion in the methodology section of the paper. We have included an explanation for why we chose this particular material for testing. By doing so, we hope to provide a clear rationale for our material selection, enhancing the comprehensibility and significance of our research..

Comment 3: It would also be nice to divide the methodology section into subsections: Carrying out mechanical tests, Sample processing, Measurement of surface parameters.

Response:

This is a very important suggestion. We have taken it into account and divided the methodology section into the subsections:  This organizational change allows for a more structured presentation of our research methodology, making it easier for readers to navigate and understand the experimental procedures.

Comment 4: Table 3 is a set of drawings and should be formatted as Figure 3 a, b, c.

Response:

We have revised the formatting of Table 3 as per your suggestion, and it is now presented as Figure.

Comment 5: In the figure you provide wear data. What is the spread of the obtained experimental values? Specify the confidence level..

Response:

Thank you for your question. We have added error bars to the figure, which indicate the spread of the obtained experimental values.

Comment 6: In figures 6 and 7, the confidence probability is quite large, so it is quite difficult to talk about an increase or decrease in cutting forces. When describing, one should be more careful with this.

Response:

Thank you for your valuable input. We wholeheartedly agree with your observation.  Consequently, when describing the results, we have exercised caution in interpreting the data, recognizing that the observed trends represent delicate tendencies rather than definitive increases or decreases in cutting forces..

Comment 7: It would also be nice to have a small diagram in the methodology section showing the measured cutting force acting on the cutting insert.

Response:

We have included a diagram in the methodology section, illustrating an example of the thrust force during cutting (Figure 4 at revised manuscript)

Comment 8: You have a large amount of experimental material but little discussion of the results obtained. It would be good to describe in more detail what the resulting results are connected with, to provide data on the interaction of the tool material with the workpiece being processed..

Response:

We have taken your suggestion into account and expanded the discussion in the article. We have tried more detailed explanations of the connections between the obtained results and the interactions of the tool material with the workpiece during the milling.

I appreciate for Editors/Reviewers warm work earnestly, and hope that the corrections will meet with approval. Once again, I thank you very much for your comments and suggestions.

Yours sincerely,

Jakub Matuszak

Kazimierz Zaleski

Andrzej Zyśko

Reviewer 2 Report

The author studied the role of cutting speed as a key factor in affecting tool wear, surface layer properties and fatigue life in the high-speed machining process of titanium alloy. The effect of cutting speed on flank wear, thrust force, cutting torque, surface roughness, surface layer microhardness, residual stress and fatigue life were obtained, and the comprehensive evaluation of machinability for titanium alloy with different cutting speeds was realized. The manuscript is not only closely related to the theme of the journal, but also has sufficient experiments and abundant characterization methods. This has important theoretical significance and application value for the reasonable determination of high-speed cutting process parameters for titanium alloy. However, in order to meet the publication requirements of Materials, the following problems need to be solved:

1. In Abstract, it is necessary to summarize the important findings in this research work to attract readers’ interest, rather than to roughly describe the research content.

2. In Line 21-22 (Page 1), keywords are not randomly listed but need to select appropriate words considering the research focus of this work.

3. In Section 1, the format of quoted literature is not standardized. For example, in Line 68-69 (Page 2), it is stated that “The authors of the article [5] focused on evaluating the fracture toughness of WC cutting inserts, specifically the SM25T (HW) tungsten carbide”. The correct format of the reported literature is usually described as “Samborski et al. [5] focused on evaluating the fracture toughness of WC cutting inserts, specifically the SM25T (HW) tungsten carbide”. Thus, it is necessary to completely modify the format of the literature reported in this part.

4. In Section 3, the order of the chapters must be checked. For example, the order of titles in Line 322 343 and 355 are same.

5. The ordinate in Fig. 8 has repeated scales and needs to be modified.

6. This manuscript lacks the content to analyze the results and needs to be supplemented as a discussion section.

7. In Section 4, it is necessary to enrich the substantive content of the conclusion, rather than simply and roughly describe some objective facts. For example, in Line 438 (Page 16), it is stated that “the cutting speed affects the thickness of the hardened surface layer”.

        8. The organization structure and grammar of the whole manuscript need to be improved and modified.

         The organization structure and grammar of the whole manuscript need to be improved and modified.

Author Response

Article: Investigation of the Impact of High-Speed Machining in the Milling Process of Titanium Alloy on Tool Wear, Surface Layer Properties, and Fatigue Life of the Machined Object

Responses to comments from Reviewer 2

I would like to thank the Editor for their consideration, and the Reviewer for the time spent on carefully reviewing this work and for their valuable deep insight and comments. I feel that this paper is now clearer, more thoroughly discussed and better-referenced.

The work has been revised to address the reviewers’ suggestions. Please find hereafter a  point-by-point reply to the comments and suggestions. Any revisions to the manuscript was marked up using the “Track Changes”.

General comment from Reviewer 2:  The author studied the role of cutting speed as a key factor in affecting tool wear, surface layer properties and fatigue life in the high-speed machining process of titanium alloy. The effect of cutting speed on flank wear, thrust force, cutting torque, surface roughness, surface layer microhardness, residual stress and fatigue life were obtained, and the comprehensive evaluation of machinability for titanium alloy with different cutting speeds was realized. The manuscript is not only closely related to the theme of the journal, but also has sufficient experiments and abundant characterization methods. This has important theoretical significance and application value for the reasonable determination of high-speed cutting process parameters for titanium alloy. However, in order to meet the publication requirements of Materials, the following problems need to be solved.

Response:

We would like to thank the Reviewer for his very good opinion. We feel to be obligated to answer for points mention in the review. The paper has been modified and improved. We believe that now is clearer.

Comment 1:  In Abstract, it is necessary to summarize the important findings in this research work to attract readers’ interest, rather than to roughly describe the research content.

Response:

We have addressed your suggestion and made the necessary changes in the Abstract. We have now provided a concise and clear summary of the important findings in this research work, aiming to capture readers' interest and emphasize the significance of our study.

Comment 2: In Line 21-22 (Page 1), keywords are not randomly listed but need to select appropriate words considering the research focus of this work.

Response:

This is a very valid point. We have revised and selected appropriate keywords that align with the research focus of this work. The updated keywords now accurately reflect the core themes and topics covered in our study.

Comment 3: In Section 1, the format of quoted literature is not standardized. For example, in Line 68-69 (Page 2), it is stated that “The authors of the article [5] focused on evaluating the fracture toughness of WC cutting inserts, specifically the SM25T (HW) tungsten carbide”. The correct format of the reported literature is usually described as “Samborski et al. [5] focused on evaluating the fracture toughness of WC cutting inserts, specifically the SM25T (HW) tungsten carbide”. Thus, it is necessary to completely modify the format of the literature reported in this part.

Response:

Thank you for pointing out the issue with the format of the quoted literature. We appreciate your attention to detail  which has led to the improvement of the paper's citation style and overall presentation. We have completely revised the format in Section 1, ensuring that all cited literature follows the standard format as suggested.

Comment 4: In Section 3, the order of the chapters must be checked. For example, the order of titles in Line 322 343 and 355 are same.

Response:

Thank you for bringing this to our attention. The error has been corrected. Section titles have been appropriately reorganized to avoid duplication and confusion.

Comment 5: The ordinate in Fig. 8 has repeated scales and needs to be modified..

Response:

Thank you for your observation. The scale in figure 8 (figure 11 in the revised manuscript) has been corrected.

Comment 6: This manuscript lacks the content to analyze the results and needs to be supplemented as a discussion section.

Response:

We have taken your suggestion into account and expanded the discussion in the article. We have tried more detailed explanations of the connections between the obtained results and the interactions of the tool material with the workpiece during the processing.

Comment 7: In Section 4, it is necessary to enrich the substantive content of the conclusion, rather than simply and roughly describe some objective facts. For example, in Line 438 (Page 16), it is stated that “the cutting speed affects the thickness of the hardened surface layer”.

Response:

We have revisited Section 4 and enriched the substantive content of the conclusion to provide more in-depth insights and analysis

Regarding the statement about the cutting speed affecting the thickness of the hardened surface layer, we have provided a more comprehensive explanation of the relationship between the cutting speed and the observed changes in the hardened surface layer thickness.

Comment 8: The organization structure and grammar of the whole manuscript need to be improved and modified.

Response:

Thank you for your suggestion. We have made efforts to improve the organization structure and grammar throughout the entire manuscript. Additionally, we have sought the assistance of a professional English language translator to check revised manuscript.

I appreciate for Editors/Reviewers warm work earnestly, and hope that the corrections will meet with approval. Once again, I thank you very much for your comments and suggestions.

Yours sincerely,

Jakub Matuszak

Kazimierz Zaleski

Andrzej Zyśko

Reviewer 3 Report

Manuscript ID: materials-2507951-peer-review-v1

Despite the efforts done by the authors, and the manuscript contains an intersting scientific story, there are some important points that should be considered by the authors.

Comments and Suggestions for Authors:

1.    Abstarct was written in a general form and does not contain any results or numbers. It should contain the aim of the work and the most important results.

2.     Please rewrite the method of citing references in a conventional way and do not talking about numbers, for eample "The study [1] presents the study"…." In the article [2]"

3.     There are some paragraphs in the introduction that need to be summarized.

4.     It is useful for the authors to write a short paragraph about the thermal properties of the Ti alloys. (https://doi.org/10.3390/ma16134768, doi: 10.1088/1757-899X/973/1/012025 "Phase stability of mechanically alloyed Ti-Fe-Al"

5.     Please mention the standard used for fatigue test specimens.

6.     In Figs. 1 and 4, the authors added a photo image and an engineering drawing. The photograph does not add anything to the scientific content.

7.     Table 1… What are the dimensions used?

8.     In Table 2, Chemical composition…All comma should be changed to dot. For example  6,25-6,31 should be changed to  6.25-6.31"  

9.     In table 2, Rm, E, and HRC are not physical properties but mechanical properties.

10.             What do you mean by "VB3"? Clarifiy all abbreviations.

11.             In the caption of Figure 4, what does Figure 4, b, refer to?

12.            Why is the surface roughness at 190 m/min the lowest value?

13.             In conclusion vc = 250÷310 m/min should be vc of (250 to 310) m/min

14.             Some grammatical mistakes are detected. The language should be improved.

 Some grammatical mistakes are detected. The language should be improved.

Author Response

Article: Investigation of the Impact of High-Speed Machining in the Milling Process of Titanium Alloy on Tool Wear, Surface Layer Properties, and Fatigue Life of the Machined Object

Responses to comments from Reviewer 3

I would like to thank the Editor for their consideration, and the Reviewer for the time spent on carefully reviewing this work and for their valuable deep insight and comments. I feel that this paper is now clearer, more thoroughly discussed and better-referenced.

The work has been revised to address the reviewers’ suggestions. Please find hereafter a  point-by-point reply to the comments and suggestions. Any revisions to the manuscript was marked up using the “Track Changes”.

General comment from Reviewer 3:  Despite the efforts done by the authors, and the manuscript contains an intersting scientific story, there are some important points that should be considered by the authors.

Response:

We would like to thank the Reviewer for valuable opinion. We feel to be obligated to answer for points mention in the review. The paper has been modified and improved. We believe that now is clearer.

Comment 1:  Abstarct was written in a general form and does not contain any results or numbers. It should contain the aim of the work and the most important results.

Response:

We have addressed your suggestion and made the necessary changes in the Abstract. We have now provided a concise and clear summary of the important findings in this research work, aiming to capture readers' interest and emphasize the significance of our study.

Comment 2: Please rewrite the method of citing references in a conventional way and do not talking about numbers, for eample "The study [1] presents the study"…." In the article [2]".

Response:

Thank you for pointing out the issue with the format of the quoted literature. We appreciate your attention to detail which has led to the improvement of the paper's citation style and overall presentation.

Comment 3: There are some paragraphs in the introduction that need to be summarized.

Response:

Thank you for your suggestion. We have completely revised the format in Section 1. We have made the necessary changes to the introduction. We have shortened the introductory and general part while ensuring that the key points are still adequately addressed. Additionally, we have reduced the level of detail regarding the authors' work, focusing only on the essential aspects.

 Comment 4: It is useful for the authors to write a short paragraph about the thermal properties of the Ti alloys. (https://doi.org/10.3390/ma16134768, doi: 10.1088/1757-899X/973/1/012025 "Phase stability of mechanically alloyed Ti-Fe-Al".

Response:

Thank you for your valuable suggestion. We have taken it into consideration and we have added a short paragraph in the manuscript that discusses the thermal properties of Ti alloys. Additionally, we have incorporated information from the cited articles in bibliography.

Comment 5: Please mention the standard used for fatigue test specimens.

Response:

In our research, we adopted our own buided workstand for these comparative investigations on the influence of cutting speeds during milling on fatigue life. The specimens were subjected to cyclic bending with a pre-determined constant deflection, allowing us to assess their fatigue behavior.

We believe that this approach of our study  gives valuable results into the fatigue life under different cutting conditions.

Comment 6: In Figs. 1 and 4, the authors added a photo image and an engineering drawing. The photograph does not add anything to the scientific content.

Response:

Thank you for your feedback. We have removed the photo from both Figures. We understand the importance of maintaining the scientific content and ensuring that all visual elements serve a clear purpose in conveying the research findings.

Comment 7: Table 1… What are the dimensions used?.

Response:

Thank you for your attention. Information about units has been added in the Figure description.

Comment 8: In Table 2, Chemical composition…All comma should be changed to dot. For example  6,25-6,31 should be changed to  6.25-6.31".

Response:

Thank you for bringing this to our attention. We have reviewed Table 2 and made the necessary changes to replace all commas with dots in the chemical composition values.

Comment 9: In table 2, Rm, E, and HRC are not physical properties but mechanical properties.

Response:

Thank you for your suggestion. The word "physical" properties has been replaced with the word "mechanical" properties in table.

Comment 10: What do you mean by "VB3"? Clarifiy all abbreviations.

Response:

Thank you for your question. In the revised manuscript, we have provided the definition of "VB3" as "localized flank wear" along with the reference to ISO 8688-2:1996 standard. We believe this addition clarifies the terminology and ensures that readers can easily understand the specific wear type being referred to in our study.

Comment 11: In the caption of Figure 4, what does Figure 4, b, refer to?.

Response:

thank you for your insightful observation, the photo in Figure 4b has been removed in the revised manuscript.

Comment 12: Why is the surface roughness at 190 m/min the lowest value?.

Response:

Thank you for your valuable attention. We expect that with the increase in cutting speed up to 190 m/min, the temperature increases, which causes heating of the material and a decrease in the friction coefficient, while above this speed, rapid edge wear occurs (Figure 8 in revised manuscript), which may be the dominant factor influencing the increase in surface roughness.

Comment 13: In conclusion vc = 250÷310 m/min should be vc of (250 to 310) m/min.

Response:

Thank you for your valuable insight, the changes have been included in the text.

Comment 14: Some grammatical mistakes are detected. The language should be improved..

Response:

Thank you for your suggestion. We have made efforts to improve the organization structure and grammar throughout the entire manuscript. Additionally, we have sought the assistance of a professional English language translator to check revised manuscript.

I appreciate for Editors/Reviewers warm work earnestly, and hope that the corrections will meet with approval. Once again, I thank you very much for your comments and suggestions.

Yours sincerely,

Jakub Matuszak

Kazimierz Zaleski

Andrzej Zyśko

Round 2

Reviewer 1 Report

The authors have finalized the article well based on the comments. The article can be published in my opinion.

Reviewer 2 Report

All my comments were fully adressed. It could be accepted in its present form.

Reviewer 3 Report

The authors have made all required modifications.